# Canonical BMP Signaling Executes Epithelial-Mesenchymal Transition Downstream of SNAIL1

**DOI:** 10.3390/cancers12041019

**Published:** 2020-04-21

**Authors:** Patrick Frey, Antoine Devisme, Monika Schrempp, Geoffroy Andrieux, Melanie Boerries, Andreas Hecht

**Affiliations:** 1Institute of Molecular Medicine and Cell Research, Faculty of Medicine, University of Freiburg, 79104 Freiburg, Germany; patrick.frey@sgbm.uni-freiburg.de (P.F.); Monika.Schrempp@mol-med.uni-freiburg.de (M.S.); 2Spemann Graduate School of Biology and Medicine (SGBM), University of Freiburg, 79104 Freiburg, Germany; 3Faculty of Biology, University of Freiburg, 79104 Freiburg, Germany; antoine.devisme@mol-med.uni-freiburg.de; 4Institute of Medical Bioinformatics and Systems Medicine, Medical Center—University of Freiburg, Faculty of Medicine, University of Freiburg, 79106 Freiburg, Germany; andrieux.geoffroy@gmail.com (G.A.); m.boerries@dkfz-heidelberg.de (M.B.); 5German Cancer Consortium (DKTK), Freiburg, University of Freiburg, 79104 Freiburg, Germany; 6German Cancer Research Center (DKFZ), 69120 Heidelberg, Germany; 7Comprehensive Cancer Center Freiburg (CCCF), Medical Center—University of Freiburg, Faculty of Medicine, University of Freiburg, 79106 Freiburg, Germany; 8BIOSS Centre for Biological Signalling Studies, University of Freiburg, 79104 Freiburg, Germany

**Keywords:** Epithelial-mesenchymal transition (EMT), SNAIL1, BMP signaling, SMAD, colorectal cancer, pancreatic cancer, invasion, metastasis

## Abstract

Epithelial-mesenchymal transition (EMT) is a pivotal process in development and disease. In carcinogenesis, various signaling pathways are known to trigger EMT by inducing the expression of EMT transcription factors (EMT-TFs) like SNAIL1, ultimately promoting invasion, metastasis and chemoresistance. However, how EMT is executed downstream of EMT-TFs is incompletely understood. Here, using human colorectal cancer (CRC) and mammary cell line models of EMT, we demonstrate that SNAIL1 critically relies on bone morphogenetic protein (BMP) signaling for EMT execution. This activity requires the transcription factor SMAD4 common to BMP/TGFβ pathways, but is TGFβ signaling-independent. Further, we define a signature of BMP-dependent genes in the EMT-transcriptome, which orchestrate EMT-induced invasiveness, and are found to be regulated in human CRC transcriptomes and in developmental EMT processes. Collectively, our findings substantially augment the knowledge of mechanistic routes whereby EMT can be effectuated, which is relevant for the conceptual understanding and therapeutic targeting of EMT processes.

## 1. Introduction

Epithelial-mesenchymal transition (EMT) is a conserved cellular program with fundamental roles in development, physiology, and various forms of disease. In the cancer field, EMT has been described to be conducive to tumor cell invasion, stemness and therapy resistance, and is therefore thought to expedite metastasis, the leading cause of cancer-related death [1]. Although originally it was proposed that the complete passage through EMT might be universally required for metastasis, several recent studies challenged such a simplistic view [2,3]. Therefore, a refined model was brought forth in which the extent of EMT and its contributions to distinct aspects of cancer progression are variable and apparently display pronounced cell-type as well as cancer-type specificity [4].

EMT typically entails the gain of motility and invasiveness at the expense of apico-basal polarity and cell–cell adhesion. Mechanistically, these cellular alterations are manifestations of massive gene expression changes, which can be triggered by a variety of extracellular signals. A central event seemingly common to all conditions that elicit EMT, is the upregulation of a group of master regulators, the so-called EMT-transcription factors (EMT-TFs), most notably comprising SNAIL, ZEB and TWIST family proteins [5,6]. These master regulators in turn orchestrate the EMT process by initiating the up- and downregulation of large cohorts of genes specifying mesenchymal and epithelial cell states, respectively [4,7]. However, while there is ample knowledge about conditions which induce EMT and EMT-TF expression, there is much less information about the signal transduction pathways and transcription factors which are engaged by EMT-TFs for the execution of EMT. In part, this may be owed to the fact that current insights into EMT-regulatory cascades are largely derived from a limited number of model systems with a preponderance of breast cancer models. Therefore, the full spectrum of EMT executioner pathways still has to be discovered. In fact, recent studies have hinted that EMT implementation in different settings involves considerable mechanistic diversity [7,8], highlighting the need to further investigate how EMT can be brought about.

Bone morphogenetic protein (BMP) signaling is a conserved branch of the transforming growth factor beta (TGFβ) superfamily of signaling pathways. BMP signaling events have central functions in development and in adult organs [9]. They are triggered when soluble ligands, the BMPs, bind to serine-threonine kinase receptors in the plasma membrane. As a result, the receptors multimerize and stimulate different routes of intracellular signaling. Canonical BMP signaling entails phosphorylation of the receptor SMADs SMAD1, SMAD5, and SMAD8 (SMAD1/5/8), which subsequently form gene-regulatory complexes with the universal co-SMAD SMAD4. Alternatively, BMPs can signal in a non-canonical fashion through a variety of intracellular kinases [10]. In cancer development, BMP signaling has been described to be both growth-promoting as well as tumor-suppressive [11]. In colorectal cancer (CRC), for example, BMP signaling was traditionally viewed as strictly tumor-suppressive [12], while more recent studies have also ascribed oncogenic roles to it [13,14,15].

Among the members of the TGFβ superfamily of growth factors, TGFβ1, TGFβ2, and TGFβ3 are well-described to function as upstream inducers of EMT by triggering the expression of EMT-TFs [10]. There is also evidence for the cooperation of TGFβ-induced SMAD complexes with EMT-TFs [16,17,18]. In contrast, BMPs are less renowned as triggers of EMT. Moreover, whether BMPs and BMP-induced SMAD activity is restricted to EMT induction, or can likewise act in EMT execution is not known. Here, by using human CRC and mammary epithelial cell line models of inducible EMT, we report that canonical, SMAD-dependent BMP signaling is necessary for the actuation of EMT downstream of SNAIL1. We define a BMP-dependent gene expression signature in the SNAIL1-induced transcriptome that contains several genes known to be critical for EMT execution. Expression of these BMP signature genes is recapitulated in human CRC transcriptomes, and has prognostic value for CRC patient survival. Furthermore, we show that BMP signature genes are also regulated during EMT in embryonic development, arguing for a widespread involvement in EMT processes. These findings substantially expand our knowledge about the mechanistic routes of EMT implementation and bear novel implications for the therapeutic targeting of EMT processes.

## 2. Results

### 2.1. SNAIL1 Activates BMP Target Genes during EMT in Colorectal Cancer Cells

Previously, we established an inducible EMT model system based on the LS174T colorectal adenocarcinoma cell line [19]. Upon addition of doxycycline (Dox), the LS174T cell derivatives overexpress murine HA-tagged Snail1 (Snail1-HA) and undergo a rapid, full-blown EMT (Figure 1a). The morphological changes are accompanied by the downregulation of epithelial markers E-CADHERIN (*CDH1*) and CLAUDIN-3 (*CLDN3*), and upregulation of mesenchymal markers FIBRONECTIN (*FN1*) and OB-CADHERIN (*CDH11*) [20] (Figure 1b,c). In order to identify pathways and factors operating in EMT execution, we analyzed time-resolved transcriptome data derived from this model [21]. Gene set enrichment analysis (GSEA) based on Consensus and GO-term databases revealed that several gene sets related to TGFβ superfamily signaling, including BMP signaling, are significantly enriched in the Snail1-HA-inducible transcriptome (Figure 1d). Furthermore, TF-signature analysis indicated that the BMP pathway-specific SMAD1 and SMAD5 proteins might act upon genes which are upregulated in the presence of Snail1-HA (Figure 1e). Additionally, there was an enrichment for TF-signatures of the BMP target gene and osteoblastogenesis master regulator RUNX2 and its interaction partner ATF-4 [22]. Upregulation of exemplary BMP target genes in the presence of Snail1-HA was confirmed by qRT-PCR (Figure 1f). These comprise the well-described canonical BMP target gene *ID1* [23] as well as several transcription factors that are regulated by BMP signaling in osteoblastic differentiation and skeletal morphogenesis (*DLX3/5*, *MSX2*, *RUNX1/2*, *SP7*) [24,25,26]. Collectively, these results point towards an involvement of BMP signaling in the implementation of Snail1-HA-induced EMT.

### 2.2. BMP Signaling is Required for Execution of Snail1-Induced EMT

The gene expression analyses described so far indicate that Snail1-HA overexpression leads to an increase in BMP pathway activity. To further demonstrate this, we examined phosphorylation of SMAD1/5/8 as a readout for the activation of canonical BMP signaling (Figure 2a). In accordance with previous reports [13], we found that LS174T cells possess an active BMP pathway already in the absence of Snail1-HA, which manifested in a basal level of SMAD1/5/8 phosphorylation (Figure 2b,c; lanes 1). This also applies to the HT29 CRC cell line (Appendix A). More importantly, SMAD1/5/8 amounts and phosphorylation levels increased after induction of Snail1-HA in both cell lines (Figure 2b,c, lanes 4; Appendix A), indicative of BMP pathway hyperactivation downstream of Snail1-HA in CRC cell lines.

To further investigate the functional contribution of BMP signaling to EMT execution, we made use of two BMP inhibitors interfering with the pathway by different mechanisms of action (Figure 2a). LDN193189 (LDN) is a small molecule inhibitor of BMPR1A/ALK3 kinase activity. Noggin is a physiological BMP antagonist that traps BMP ligands extracellularly, thereby preventing them from receptor binding and pathway activation. Initial tests were conducted to optimize inhibitor concentration and to monitor the time course of inhibitor action (Appendix A). When applied at the respective concentrations found to maximally reduce SMAD1/5/8 phosphorylation, LDN was observed to take full effect already after ≤ 1 h, while Noggin required ≥ 3 h of exposure time for complete pathway inhibition. This, however, is compatible with its mechanism of action. Furthermore, both inhibitors blocked the expression of ID1 (Appendix A). Additionally, LDN abolished activity of a BMP signaling reporter gene construct (Appendix A).

Having demonstrated the efficacy of both inhibitors, we next assessed the functional importance of BMP signaling in Snail1-HA-induced EMT. When Dox-induced LS174T-Snail1-HA and HT29-Snail1-HA cells were additionally treated with the BMP inhibitors, we found that the EMT-associated SMAD1/5/8 phosphorylation was diminished (Figure 2b,c and Appendix A), and the upregulation of ID1 was reduced in LS174T-Snail1-HA cells (Figure 2d). Importantly, BMP inhibition (BMPi) did not interfere with Snail1-HA overexpression.

Next, we inspected the impact of BMPi on several cellular features which are typically affected by EMT. First, we examined the morphological conversion accompanying EMT. In control conditions, Snail1-HA-expressing LS174T cell clusters completely dissipated and cells assumed an elongated morphology after 72 h of Dox administration (Figure 2e). In contrast, under BMPi, while there was still a noticeable change upon Dox addition compared to the uninduced state, cells failed to disperse and to acquire an elongated mesenchymal phenotype. Instead, they remained more clustered and epithelial (Figure 2e). Importantly, in the absence of Snail1-HA, BMPi had no effect on cellular morphology (Appendix A). Next, we tested whether BMPi had an effect on EMT-induced anoikis resistance [27]. However, perhaps not surprisingly for adenocarcinoma cells, we observed LS174T cells to be anoikis-resistant, irrespective of the presence or absence of Snail1-HA and BMPi (Appendix A). In contrast, BMPi abrogated the sprouting behavior acquired after Snail1-HA overexpression in a three-dimensional spheroid invasion assay (Figure 2f). Again, Snail1-HA-deficient spheroids were not discernibly affected by BMPi (Appendix A). Moreover, the inhibitory effect of BMPi on the EMT process was also observed when cells were treated for longer periods of time (Appendix A), indicating that BMPi does not simply delay the EMT process, but rather imposes a permanent roadblock. Unfortunately, corresponding analyses were not informative for HT29-Snail1-HA cells, because these did not discernibly undergo EMT and showed no response to BMPi under the experimental conditions (Appendix A; see below, Section 2.4.).

Previous reports suggested a role for TGFβ1-induced SMAD-complexes in mediating EMT-TF function [16,18], and several gene sets related to TGFβ signaling were found to be enriched in the Snail1-HA-induced transcriptome of LS174T cells (Figure 1d). Despite this, it is highly unlikely that TGFβ signaling and TGFβ pathway specific-SMADs contribute to the Snail1-HA-induced EMT process in addition to the BMP pathway. First, LS174T cells carry a homozygous 1 bp deletion in exon 3 of the TGFβ-receptor type 2 gene (*TGFBR2*) [28,29]. Second, by investigating SMAD2/3 phosphorylation, we confirmed that LS174T do not have a functional TGFβ signaling pathway (Appendix A). Third, inhibition of TGFβ signaling using the TGFBR1A/ALK5 inhibitor SB431542 did not affect the regulation of EMT marker genes (Appendix A) and the morphological changes observed during EMT (Appendix A). Thus, the effects observed by BMPi cannot be recapitulated by TGFβ pathway blockade, and we rule out a contribution of TGFβ signaling to Snail1-HA-induced EMT execution in LS174T cells.

### 2.3. Transcriptome Analyses Define a BMP-Dependent EMT Gene Signature

Next, we sought to comprehensively characterize the impact of BMP pathway inhibition on gene regulation downstream of Snail1-HA. Therefore, we performed microarray-based, time-resolved transcriptome profiling of the EMT process under control conditions and under BMPi. Principal component analysis (PCA) of the analyzed samples showed high similarity between the biological replicates, as well as a clear separation of the uninduced and Dox-induced samples over time along the PC1 axis (Appendix A). Furthermore, after 72 h of induction, BMPi samples separated from controls along the PC1 axis, which we assume to reflect the impaired EMT process observed under BMPi. We next determined the genes whose regulation by Snail1-HA was significantly perturbed under BMPi (Figure 3a and Appendix A and Appendix A). Although after 6 h of Dox treatment, no Snail1-HA-regulated genes were significantly altered by BMPi, we found 24 and 653 genes to be impaired in their regulation after 24 h and 72 h of BMPi, respectively. We defined the group of genes that were affected in their Snail1-HA-dependent regulation both by Noggin and LDN193189 at least at one time point, to be BMP-dependent with high confidence, and termed them BMP signature genes. These amount to 282 genes, corresponding roughly to one twentieth of all Snail1-HA-regulated genes. Of the 282 BMP signature genes, 208 genes were upregulated and 74 genes downregulated after Snail1-HA-induction. By GSEA, we found the BMP signature to be significantly enriched for genes implicated in mesenchymal and epithelial differentiation, as well as in cell adhesion and migration (Figure 3b and Appendix A). Furthermore, the BMP signature genes showed significant enrichment for GO-terms directly associated with EMT, as well as for a previously defined EMT core gene list [30] (Figure 3c and Appendix A). Notably, among the 18 genes shared by the BMP signature and the enriched EMT-related gene sets, there were several genes which are well known to be downregulated (*CDH1*, *FOXA1*, *FOXA2*) and upregulated (*CDH11*, *LEF1*) across several EMT models, and which perform critical functions during EMT (Appendix A). The microarray-derived expression changes for these genes, the epithelial marker *CLDN3*, and additionally *RUNX2* are visualized in Figure 3d and Appendix A. The heatmaps show that regulation of *CDH1*, *FOXA1*, *FOXA2*, *CDH11*, and *LEF1* by Snail1-HA was attenuated by BMPi. *RUNX2* upregulation was mitigated by LDN only. Repression of *CLDN3*, which is not a BMP signature gene and served as control, was unaffected (Figure 3d).

To validate the results of the microarray studies and to confirm the impact of BMPi on the selection of EMT-relevant genes, we analyzed their expression on protein and RNA level. Additionally, we investigated the EMT markers *FN1* and *ZEB1*, which we had not been able to monitor in the microarray experiments probably due to technical issues with the probes, and the epithelial maintenance factor and ZEB1 antagonist *GRHL2* [31,32,33]. Altogether, we could verify that downregulation of E-CADHERIN (*CDH*1), FOXA1, and FOXA2 was impaired by BMPi (Figure 3e and Appendix A). Likewise, we corroborated that upregulation of OB-CADHERIN (*CDH11*), LEF1 and RUNX2 by Snail1-HA was prevented by BMPi. This also applies to FIBRONECTIN (*FN1*). In contrast, regulation of CLAUDIN-3 (*CLDN3*), GRHL2, and ZEB1 by Snail1-HA did not depend on BMP signaling. However, we have shown before that ZEB1 is not required for EMT in our model system [34], and persistent upregulation of ZEB1 could explain why *GRHL2* repression was not prevented by BMPi [31,32,33]. In summary, we could define a high-confidence BMP-dependent gene expression signature in the Snail1-HA-regulated EMT transcriptome.

The signature is enriched for EMT-related gene sets and contains several genes previously associated with EMT. Furthermore, it compulsorily contains genes critical for morphological conversion and invasion acquired during EMT, because both processes are blocked by BMPi. Therefore, although the BMP-dependent gene signature represents only a small fraction of Snail1-HA-induced gene expression changes, it is valuable for the identification of functionally important genes in EMT execution.

### 2.4. Knockout of SMAD4 Blocks BMP-Dependent EMT Execution

After having characterized the effects of BMPi on the level of gene regulation, we strived to obtain insight into the mechanism whereby BMP signaling contributes to Snail1-HA-induced EMT. Since TGFβ superfamily signaling pathways can relay signals intracellularly by different routes (Figure 2a), an important question was whether the observed effects are mediated by canonical BMP signaling or involve alternative factors. To investigate this, we used CRISPR/Cas9 genome editing to knock out *SMAD4* in LS174T cells. SMAD4 is the universal co-SMAD used in TGFβ superfamily signaling and is typically required for SMAD-mediated signal transduction. By using two gRNAs in flanking intronic regions, we deleted exon 9 of the *SMAD4* gene (Appendix A). This exon was previously targeted by others to disrupt *SMAD4* in human and murine systems, and its deletion leads to complete protein loss [35,36,37,38]. By single-cell sorting and PCR-based deletion screening, we selected three knockout (ko) and two wild-type (wt) cell clones (Appendix A). In analogy to the parental EMT model system, we reconstituted the *SMAD4* wt and ko clones with the overexpressing vector for Snail1-HA and the corresponding control. As expected, the ko clones did not show any detectable SMAD4 on protein level (Figure 4a) and displayed drastically reduced SMAD4 RNA levels (Figure 4b), probably as a result of nonsense-mediated decay. Correspondingly, they lost expression of the canonical BMP signaling target *ID1*, demonstrating that *SMAD4* ko completely abrogated basal SMAD-mediated BMP signaling in LS174T.

When we treated the cells with Dox to induce Snail1-HA expression, we observed that the *SMAD4* ko clones displayed a strongly impaired EMT phenotype, highly reminiscent of what we had observed with BMPi (Figure 4c and Appendix A). Likewise, in contrast to the SMAD4 wt clones, they did not show any invasive capacity (Figure 4d). This was not due to the reduced spheroid size of the knockout clones, as they also failed to form invasive sprouts after compensating in size after increased periods of time (Appendix A). We then asked how *SMAD4* ko affected the expression of EMT-related genes (Figure 4e and Appendix A). In doing so, we observed that OB-CADHERIN (*CDH11*) was expressed at higher basal levels in the knockout clones #58 and #65. Similarly, FIBRONECTIN (*FN1)* expression was elevated in clone #58 on protein level. These alterations are probably due to clonal variability. More importantly, however, Snail1-HA induction led to an upregulation of OB-CADHERIN (*CDH11*) only in the parental cells and in the wt clone #11, but in none of the three *SMAD4* ko clones. Likewise, although *FN1* RNA expression was also induced in the knockout clones #63 and #65, FIBRONECTIN protein levels were only increased in the parental cells and the wt clone #11. Similarly, the epithelial marker E-CADHERIN (*CDH1*) was only downregulated in cells with wt *SMAD4*. Further, *SMAD4* ko completely blocked the induction of LEF1 and RUNX2, and prevented the downregulation of FOXA1 and FOXA2 (Figure 4e and Appendix A). On the contrary, the BMP-insensitive genes *CLDN3* and *ZEB1* showed down- and upregulation, respectively, regardless of *SMAD4* status. Of note, the wt clone #30 showed only attenuated regulation of EMT-associated genes, which is probably due to impaired Snail1-HA inducibility in this cell clone. Nevertheless, it tendentially recapitulated the expression changes observed in the parental cells and in wt clone #11.

In their sum, these findings prove that SMAD4 is required for EMT execution downstream of Snail1-HA in the LS174T model system. The observed SMAD4-dependence could also explain the different behavior of HT29 cells in response to Snail1-HA expression and BMPi (see Section 2.2.; Appendix A), because HT29 cells have a defective *SMAD4* gene [28]. Moreover, since the *SMAD4* ko closely phenocopies the effects observed using the BMP antagonist and receptor inhibition, Snail1-HA-induced EMT seemingly does not rely on non-canonical BMP signaling mechanisms, but is mediated largely, if not exclusively, via BMP-stimulated SMAD activity.

### 2.5. BMP Signature Gene Expression Patterns are Recapitulated in Human Colorectal Cancer Tumors and Can Predict Patient Survival

Next, we asked whether the EMT-associated regulation of BMP signature genes, which we identified in vitro, could also be detected in human cancer transcriptomes. Since the BMP signature was defined in a colorectal cancer model of EMT and EMT does occur in a subset of colorectal tumors, we focused on publicly available CRC transcriptome data from the Cancer Genome Atlas (TCGA). Our BMP gene expression signature comprises two subsets of genes, which are up- or downregulated in the presence of Snail1-HA. Therefore, if this pattern of BMP signature gene regulation is conserved in tumors, we reasoned that we should be able to detect positively correlated expression among constituents of each of the two subsets of BMP signature genes, whereas the two subsets should exhibit anti-correlated expression. However, when we analyzed the correlation of gene expression in all 433 TCGA tumor samples in our cohort and applied hierarchical clustering, a highly complex pattern of correlated and anti-correlated gene expression was observed (Figure 5a). Nevertheless, there was a tendential separation of up- and downregulated BMP signature genes. We hypothesized that this incomplete segregation could be a result of the non-stratified analysis of all tumor samples, since EMT does not occur universally in CRC but is mainly limited to tumors classified as CMS4 [39]. In addition, a considerable share of tumors might not be able to recapitulate expression patterns of the BMP signature because of mutations in *SMAD4* [40]. Thus, we refined the analysis, only considering tumors that do not carry *SMAD4* mutations and classify as CMS4 (Figure 5b). This stratification yielded two clearly detectable, separate clusters of genes with positively correlated expression, and decidedly increased the anti-correlation between up- and downregulated BMP signature genes. As expected, this improvement was not observed when considering only CMS1, CMS2, or CMS3 tumors, all of which rather blurred the anti-correlation (Appendix A). These results show that CRC transcriptomes recapitulate the expression pattern of BMP signature genes induced by Snail1-HA, specifically of those from tumors showing evidence of EMT.

As EMT is known to increase tumor malignancy and metastasis, we wondered whether expression of BMP signature genes in CRC tumors carries prognostic value for CRC patient survival. We therefore divided the 433 tumor samples into two groups by k-means clustering based on the expression of the 282 BMP signature genes (Appendix A). When analyzing the survival probability of the two groups, we observed a tendential but non-significant difference (Figure 5c), which could be due to at least two different confounding factors. Again, a predictive potential of BMP signature gene expression might only be observable in a specific subtype of tumor, in this case those that are representative of our model-of-origin. Accordingly, in a first step we limited the analysis to all tumors that classify to the same CMS as LS174T cells (CMS3) [41], and do not carry *SMAD4* mutations. This stratification resulted in two groups of samples with different patterns of BMP signature gene expression (Appendix A), which showed differential survival probability (Figure 5d). Remarkably, this difference was now precisely at the threshold of statistical significance, even though the stratification drastically reduced the case number to 44 tumors. As a second reason for the weak predictive value of BMP signature gene expression, we hypothesized that not all BMP signature genes might be relevant for patient survival. Thus, further refinement of the genes under consideration to only those presumably particularly critical for EMT might increase prognostic power. We therefore limited our analysis to the 18 genes shared between the BMP signature and other EMT gene sets (see Figure 3c and Appendix A). The compound expression patterns of these 18 genes separated the cohort of 433 tumors into two groups of specimens (Appendix A), which indeed exhibited a significant difference in survival probability (Figure 5e). Taken together, it appears that the expression patterns of BMP signature genes are recapitulated in human colorectal tumors, specifically in those that underwent EMT, and that the expression of genes in the BMP signature can have prognostic value for CRC patients.

### 2.6. BMP Signaling also Contributes to EMT Execution in Other Cell Culture Models and In Vivo

After having shown the involvement of BMP signaling and BMP signature genes in a colorectal model of EMT and in human colorectal tumors, we wanted to assess whether similar mechanisms operate in other instances of EMT. For this, we turned to a widely used inducible in vitro model system of EMT based on the non-transformed human breast epithelial cell line MCF10A. These cells were engineered to overexpress SNAIL1 coupled to an estrogen receptor (ER) hormone-binding domain, thereby enabling inducible regulation of SNAIL1-ER activity by addition of 4-hydroxytamoxifen (4OH-T) [42]. After 10 days of 4OH-T administration, MCF10A-SNAIL1-ER, but not MCF10A-control cells, transitioned to a mesenchymal phenotype with cell clusters dispersing and cells becoming more elongated (Figure 6a). Analogous to LS174T-Snail1-HA, this morphological conversion was markedly, although less strongly, impaired when we simultaneously inhibited BMP signaling with LDN (Figure 6a). Moreover, by immunoblotting, we observed a similar increase in SMAD1/5/8 phosphorylation during the EMT process, which was abrogated by LDN (Figure 6b).

At the gene expression level, ZEB1, FIBRONECTIN (*FN1*), as well as the BMP signature genes DKK1 and ID2 were induced after SNAIL1-ER induction, while E-CADHERIN (*CDH1*) and FOXA1 were repressed (Figure 6b and Appendix A). *CDH11*, *CLDN3*, *GRHL2*, *LEF1* and *RUNX2* were not deregulated upon SNAIL1-ER induction. Similar to LS174T-Snail1-HA cells, induction of ZEB1 was not affected by BMPi, while upregulation of FIBRONECTIN (*FN1*), DKK1 and ID2 was impaired by BMPi. However, unlike in the LS174T cell model, downregulation of E-CADHERIN (*CDH1*) and FOXA1 by SNAIL1-ER was largely insensitive to concomitant LDN treatment (Figure 6b and Appendix A). Thus, genes upregulated during EMT in MCF10A appear to be more sensitive to BMPi than genes that become repressed. To assess whether BMPi also impairs the EMT-induced invasive behavior in the MCF10A-SNAIL1-ER model, we performed spheroid invasion assays. We found that, in contrast to LS174T, MCF10A can collectively move through collagen I irrespective of SNAIL1-ER activity (Appendix A). This type of movement produces long strands of cells radiating from the center of the spheroids. On top of that, SNAIL1-ER activation markedly increased the potential of cells to leave the cell clusters and invade the matrix individually (Figure 6c and Appendix A). When we quantified this SNAIL1-ER-enhanced single cell invasiveness, BMPi was found to significantly reduce the number of individual cells that delaminated from the strands compared to control conditions. Overall, these results resemble fairly well our observations made with LS174T-Snail1-HA cells, and argue that BMP pathway activity is engaged for the implementation of EMT downstream of SNAIL1 also in MCF10A cells.

Additionally, we searched for evidence of regulation of our BMP gene expression signature in a published dataset from a previous study which had investigated SMAD1/5-dependent gene expression in the NMuMG mouse mammary gland cell line model for TGFβ1-inducible EMT [43]. Of note, in this study, the *Snai1* gene was among the SMAD1/5-regulated genes downstream of TGFβ1. Thus, effects of SMAD1/5 activity upstream of Snail1 could not be discriminated from those downstream of Snail1. Nevertheless, we found a significant overlap of 42 genes between the SMAD1/5-dependent genes from TGFβ1-treated in NMuMG cells and the BMP signature genes from LS174T-Snail1-HA cells (Figure 6d). In sum, these results show that the regulation of BMP signature gene expression is not limited to colorectal cancer, but seemingly also contributes to EMT execution in other in vitro models of EMT based on mammary epithelial cell lines.

To further corroborate the functional importance of BMP signaling in EMT execution, we next asked whether BMP pathway activity was critical not only for the acute implementation of EMT but also to sustain mesenchymal features. To this end, we turned to PANC-1 pancreatic cancer cells which have a known mesenchymal character [44,45,46], and express the EMT-TFs SNAIL1 and ZEB1 (Figure 7a,b). Moreover, PANC-1 cells have an active BMP pathway, as indicated by readily detectable SMAD1/5/8 phosporylation (Figure 7a). When exposed to LDN, SMAD1/5/8 phosphorylation was abolished, and the BMP signature genes ID1 and ID2 were downregulated, indicative of effective BMP pathway inhibition (Figure 7a). Importantly, while the expression of SNAIL1 and ZEB1 was not affected by LDN, we observed the upregulation of the epithelial markers and BMP signature genes *CDH1* and *FOXA1* even though this increase in expression manifested only at the RNA level (Figure 7a,b). Nonetheless, BMP pathway inhibition caused PANC-1 cells to flatten and to form more compact cell clusters, further indicating a transition to a more epithelial state under BMPi (Figure 7c). Altogether, these observations argue that active BMP signaling also plays a role in maintaining a mesenchymal cell state. In agreement with the proposed function of BMP signaling in EMT execution, it thus appears that unabated expression of EMT-TFs does not suffice to uphold a fully mesenchymal state when BMP pathway activity is impaired.

Finally, we wanted to determine whether BMP signature genes are also regulated in different organisms in processes where EMT plays essential roles in vivo. To do so, we made use of two publicly available sets of genes expressed by cells in the primitive streak area during mouse gastrulation [47,48] and two gene signatures of cells at the invasive front of the cranial neural crest in the chicken [49,50]. Interestingly, in all four gene sets investigated, we found significant enrichment of the subset of BMP signature genes which is upregulated by Snail1-HA in LS174T cells (Figure 7d). This again demonstrates that the regulation of BMP signature genes during EMT is not limited to cell culture models and could play a role during developmental EMT in different organisms in vivo.

## 3. Discussion

EMT is a pivotal process in development and disease. In particular, its contributions to cancer progression and metastasis endow EMT with high clinical relevance and render it an attractive target for therapeutic intervention. Devising strategies to interfere with EMT, in turn, requires profound knowledge about the various pathways and molecular events whereby EMT can be induced and how it unfolds. In this regard, especially the signaling pathways orchestrating EMT downstream of EMT-TFs—the aspect which we termed EMT execution—are only poorly understood [4,7]. Consequently, we investigated EMT execution using CRC and mammary gland models of SNAIL1-induced EMT. We found that the BMP pathway is activated during EMT and, by using different means of pathway interference, we demonstrate that the canonical branch of BMP signaling is crucial for the implementation of a mesenchymal cell state in CRC and mammary epithelial cells, and its maintenance in pancreatic cancer cells. Moreover, a BMP-dependent gene expression pattern specifically associated with EMT was found to be recapitulated in human CRC tumors and in developmental contexts of EMT. A model for the proposed role of BMP signaling in EMT execution is shown in Figure 7e. It should be kept in mind, though, that not all aspects of EMT execution appear to be affected by BMP signaling and that some features of EMT may be regulated by SNAIL1 and other EMT-TFs in a BMP pathway-independent fashion.

Several groups have investigated connections between TGFβ superfamily signaling pathways and EMT before. However, in contrast to TGFβ signaling, which is uniformly regarded as an EMT-driving pathway, there are conflicting reports about the influence of BMP signaling on EMT processes (reviewed in [51]). While EMT-inducing functions of BMP pathway activity were described [52,53], several other groups reported BMP signaling to counteract EMT and to foster the reverse process of mesenchymal-epithelial transition [54,55,56]. Given the variety of model systems that were used for these investigations, the contrasting observations argue for context-dependent, adverse as well as favorable roles of BMP signaling in EMT processes. Importantly, however, all previous reports studied BMP signaling for its role upstream of the activation of EMT-TFs and, therefore, could not discriminate between its functions in EMT induction versus execution. For example, a recent study established functional relevance for SMAD1/5 complexes in TGFβ-driven EMT in a mouse mammary gland cell line model [43] and assigned an EMT-promoting role to BMP-associated signaling mediators. However, since during EMT the upregulation of Snail1 itself was impaired by *SMAD1/5* knockout, these findings do not allow for conclusions about potential roles of SMAD1/5 also downstream of Snail1. Hence, our study provides the first firm evidence for a specific requirement of canonical BMP-signaling in EMT execution downstream of EMT-TFs.

In colorectal carcinogenesis, both TGFβ and BMP signaling are generally considered to exert tumor-suppressive functions, for example by acting as growth-inhibitory and differentiation-promoting pathways (reviewed in [12,57]. Due to its tumor-suppressive physiological functions in the colon, it therefore may seem counterintuitive that BMP pathway activity should contribute to a process that enhances malignancy. However, several reports also demonstrated that a subset of colorectal tumors retains an active BMP pathway [58,59]. Consistent with this, we find a high basal level of BMP signaling in LS174T cells, which has also been noticed before [13]. Therefore, similar to TGFβ signaling, which is well known to undergo a switch from a tumor-suppressive to a tumor-promoting pathway in the course of oncogenic transformation [10,60], it is conceivable that BMP signaling is also subjected to a shift in function and contributes to CRC progression and metastasis.

An unresolved question at this point concerns the mechanism of how BMP signaling is harnessed by SNAIL1 for EMT execution. The BMP pathway is in the “*on*” state in LS174T and HT29 cells, but its activity seemingly increases and its transcriptional output changes in the presence of Snail1-HA. Accordingly, SNAIL1 may rely on both hyperactivation of the BMP pathway and redirection of its activity to execute EMT, although at this point we cannot disentangle the individual contributions of both. From a mechanistic point of view, however, there are potential explanations for the EMT-associated changes to BMP signaling. Both basal and Snail1-HA-hyperactivated BMP pathway activities in LS174T and HT29 are sensitive to Noggin, arguing for ligand-dependent, paracrine or autocrine activation of BMP signaling. In agreement with this, LS174T cells endogenously express BMP ligands. Notably, we found both *BMP2* and *BMP4* to be upregulated in the presence of Snail1-HA (see Appendix A), which might account for pathway hyperactivation. As for the redirection of BMP pathway-mediated transcription, it was previously reported that EMT-TFs can form complexes with TGFβ-activated SMAD proteins to regulate target genes during EMT [16,17]. Since we have shown that in LS174T cells SMAD4, and by inference SMAD1/5/8, contribute to EMT execution, it is conceivable that SNAIL1 similarly engages in complexes with BMP-activated SMADs to control gene expression. Supporting this idea, an interaction of SNAIL1 with SMAD1 and SMAD4 was recently shown in glioblastoma [61]. Furthermore, among the EMT-associated genes which are regulated in a BMP pathway-dependent manner, there are several direct targets of SNAIL1, like *CDH1*, *MYB*, and *FOXA1* [62,63,64]. This could constitute another hint that SMADs directly participate in SNAIL1-mediated gene regulation.

Hitherto, SNAIL1 is mainly described to be a transcriptional repressor [1]. Similarly, complexes formed by SNAIL1 and SMAD proteins so far were only reported to repress genes. However, the majority of genes dependent on BMP signaling during EMT in LS174T were genes upregulated in the presence of Snail1-HA (Appendix A). BMP-activated SMAD complexes could therefore be directly or indirectly employed by SNAIL1 in order to exert gene-activating functions. Further experiments will have to reveal at which stages and in which ways BMP-induced SMAD complexes partake in gene repression and gene activation aspects of EMT execution.

Regarding the functional impact of BMP signaling on the EMT process, we established a signature of 282 genes whose deregulation by Snail1-HA was dependent on BMP signaling. This signature comprises a significant number of genes with well-established and conserved roles in EMT and cell invasion. For instance, among the BMP-dependent genes that are upregulated in the presence of Snail1-HA, we find *CDH11*, *FN1*, *ID1*, *ID2*, *RUNX2*, *BAMBI*, and *LEF1*. *CDH11* and *FN1* are known mesenchymal markers with pro-invasive potential [65,66]. *ID1*, *ID2* and *RUNX2* were also shown to conduce to cancer cell invasiveness [67,68,69], and *BAMBI* is a BMP target gene implicated in CRC metastasis [70]. Similarly, we and others reported *LEF1* to be a regulator of *FN1* and critical mediator of EMT downstream of SNAIL1 [21,71]. Complementary, BMP signature genes which are downregulated in the presence of Snail1-HA include the epithelial gatekeeper genes *CDH1*, *FOXA1*, *FOXA2* and *OVOL2* [20,64,72]. Given that the BMP signature contains genes that carry out pivotal roles in EMT, it is not surprising that blocking their regulation by BMPi was sufficient to abrogate the invasiveness of LS174T-Snail1-HA cells, thereby emphasizing the importance of BMP signaling as a crucial factor in EMT execution.

Compared to the total number of genes which are deregulated during EMT, the BMP signature genes account for only a small fraction of gene expression changes. In fact, the majority of genes deregulated in the presence of Snail1-HA were not affected by BMPi. As proposed by us and others [8,21], this observation supports a model which suggests that EMT represents a hierarchical process and integrates multiple subprograms, which account for different aspects of EMT and whose regulation depends on different intermediary effectors. In the context of such a hierarchical model, we propose that the BMP pathway represents an intermediary effector which occupies a position close to SNAIL1 and which is implicated specifically in a pro-invasive EMT subprogram (Figure 7e). This view is derived from the observation that the BMP signature contains several genes that were shown to be direct target genes of SNAIL1, like *CDH1*, *MYB*, and *FOXA1* [62,63,64]. BMP signaling thus seems to affect the very first steps of the EMT gene-regulatory cascade immediately downstream of or in cooperation with SNAIL1. Moreover, for several EMT effector genes, it is still unclear how they can be upregulated during EMT. Our results close this gap by identifying BMP signaling as a missing link in the gene-regulatory cascade between EMT-TFs and EMT effectors like *RUNX2*, *LEF1* and *FN1*.

In addition to colorectal cancer cells, we discovered BMP signaling to also be engaged in EMT execution in the mammary epithelial cell line MCF10A and the pancreatic cancer cell line PANC-1. However, the impact of BMPi on the EMT phenotype and gene regulation, especially on the repression of genes during EMT, was less pronounced than in LS174T cells. On the one hand, these findings hint that BMP-activated SMADs are implicated in EMT processes more generally. On the other hand, they also suggest that the extent to which BMP pathway activity contributes to EMT execution varies in a cell-type-specific manner. Interestingly, in contrast to LS174T cells, MCF10A and PANC-1 cells carry a functional TGFβ signaling pathway, although their basal level of activation appears to be rather low [45,46,73]. Nonetheless, it is therefore conceivable that in MCF10A and PANC-1 cells, the EMT-specific functions of BMP-activated SMADs are taken over by TGFβ-induced SMAD complexes when BMP signaling is blocked, and possibly vice versa. This would argue for some redundancy among EMT executioner pathways in different settings, as was proposed previously [7,8].

When analyzing human CRC tumor transcriptomes without prior stratification, we found the pattern of BMP signature gene expression to be conserved only tendentially. However, corresponding to our in vitro observations that BMP signature gene expression is established during EMT in a SMAD4-dependent manner, we find the pattern to be considerably better recapitulated in tumors that classify to the EMT-associated subtype CMS4 and are *SMAD4* wt. Thus, the BMP signature gene expression pattern was particularly evident in tumors that most likely had undergone EMT (CMS4) and, therefore, had established BMP signature gene expression during their genesis. In contrast, we observed that the prognostic power of BMP signature gene expression increased when we stratified for CMS3 tumors. These present with the most epithelial appearance [39], and correspond to the subtype from which LS174T cells apparently originate [41]. The increase in predictive value in the CMS3 subgroup, therefore, can be explained by interpreting the extent of BMP signature gene expression as a measure of incipient EMT and a shift towards CMS4. Notably, CMS4 tumors show the worst prognosis of all four subtypes [39]. Thus, BMP signature gene expression is predictive for survival in highly epithelial tumors and best-recovered in highly mesenchymal tumors.

Aside from the cancer models, we also find evidence for BMP signature gene expression in transcriptomes from the gastrulating mouse embryo and neural crest cell invasion in chicken. Interestingly, in both systems, BMP signaling is known to be required for an EMT process to occur [74,75]. A priori, this would not allow to distinguish between contributions of BMP signaling to EMT induction versus execution. However, in mouse gastrulation, *Bmpr1a* ablation impairs the EMT-associated outcomes without affecting the expression of *Snai1* [76]. Therefore, it may well be that the EMT-executing function of BMP signaling which we discovered in a cancer context applies more broadly and also occurs in developmental instances of EMT.

In summary, our study demonstrates the employment of canonical BMP signaling in EMT execution which seems to be a feature of EMT with wide-ranging incidence. Further experiments are needed to uncover the mechanistic details of context-dependent integration of BMP signaling into EMT-associated gene regulatory networks. Independently of this, our findings suggest that BMP pathway activity could be a novel intervention point for the targeting of EMT processes.

## 4. Materials and Methods

### 4.1. Cell Culture and Generation of Stable Cell Lines

The LS174T and HT29 cell lines were obtained from the CLS Cell Line Service (#300392 and #300215, Eppelheim, Germany). MCF10A cells were ordered from ATCC (CRL-10317, Manassas, VA, USA). For all three, their identity and purity was authenticated by SNP-profiling at Multiplexion Inc. (Friedrichshafen, Germany). PANC-1 cells were obtained from Thomas Brabletz (Friedrich-Alexander-Universität Erlangen-Nürnberg, Germany) and were authenticated by STR-profiling at Microsynth AG (Balgach, Switzerland). All cell lines were incubated at 37 °C and 5% CO_2_, and were routinely screened to be free of mycoplasma infection using MycoSensor PCR Assay Kit (#302109, Santa Clara, CA, USA) according to manufacturer’s instructions. LS174T, HT29 and PANC-1 cells were cultivated in Dulbecco’s modified Eagle’s medium (DMEM) supplemented with 10% (v/v) FCS, 0.01 M HEPES/KOH, 1× MEM non-essential amino acids solution and 1× penicillin/streptomycin. LS174T parental cells and single cell clones with inducible expression constructs were produced by retroviral transduction using the pRetroX-Tight-Pur vector system as described [19]. In Figure 1 and Figure 2, and Appendix A, cells overexpressing a firefly luciferase (Luc) were used as control cells. In all other figures, cells harboring a vector without any cDNA insert (empty) were used as controls. The generation of HT29-ctrl cells and HT29-Snail1-HA clones with robust induction of Snail1-HA was described before [21]. For overexpression, LS174T and HT29 cells were treated with 0.1 µg·mL^−1^ and 1 µg·mL^−1^ Dox, respectively. MCF10A cells were kept in Gibco™ Advanced DMEM/F-12 medium, supplemented with 5% (v/v) horse serum, 20 ng/mL hEGF, 0.1 µg/mL cholera toxin, 0.5 µg/mL hydrocortisone, 1 µg/mL human insulin, and 1× penicillin/streptomycin. For constitutive overexpression of the SNAIL1-ER fusion construct, cells were retrovirally transduced with a pWZL-Blast-SNAIL1-ER-plasmid (a gift from Bob Weinberg; Addgene plasmid #18798 [77]; RRID: Addgene_18798). To generate MCF10A-ctrl cells, we used a control vector that we constructed by replacing the coding sequence of SNAIL1-ER by that of a hygromycin B resistance gene. For induction of SNAIL1-ER activity, 4OH-T was routinely administered at 100 nM. Whenever cells were treated with substances, media were refreshed every 48 h. Murine Noggin was purchased from Peprotech (#250-38, Rocky Hill, NJ, USA) and used at 100 ng mL^−1^ unless otherwise indicated. In all experiments except for the ones shown in Figure 2 and Appendix A, DMSO was added to the Noggin treated cells, to ensure comparability with the control sample. LDN193189 was obtained from Cayman Chemical (#1062368-24-4; Ann Arbor, MI, USA) and, if not pointed out otherwise, routinely applied at 50 nM. SB431542 from Selleckchem (#S1067; Houston, TX, USA) was used at 10 µM.

### 4.2. Genome Editing

The sgRNAs for CRISPR/Cas9-mediated SMAD4 knockout were chosen using the online design tools CHOPCHOP (https://chopchop.cbu.uib.no/) [78] and CCTop (https://crispr.cos.uni-heidelberg.de/) [79]. The sequences of the sgRNAs used are: intron 8-upstream-gRNA: 5′-CCTTATATCTTTCTCATGGG-3′, intron 9-downstream-gRNA: 5′-AGAACACATATAATGTACAT-3′. They were cloned into the pMuLE_ENTR_U6_stuffer_sgRNA_scaffold_R4-R3 vector (a gift from Ian Frew; Addgene plasmid # 62131 [80]; RRID: Addgene_62131) using the BfuAI restriction sites. The Cas9 enzyme was expressed from a Cas9-RFP plasmid that we described previously [21]. All newly generated plasmids were sequence-verified.

For genome editing, 2 × 10^6^ cells were nucleofected with both sgRNA-expression plasmids and the Cas9-RFP vector (1.3 µg each) using the Cell Line Nucleofector kit L (#VCA-1005, Lonza, Basel, Switzerland). Subsequently, RFP^+^ single cells were sorted into 96-well plates by FACS 72 h after nucleofection. After expansion, clones were analyzed by low-input PCR using a protocol established at the ES cell targeting core laboratory, John Hopkins University School of Medicine, Baltimore, MD, USA [81]. In brief, a minimum of around 15,000 cells were lysed in 20 µL of lysis buffer (67 mM Tris HCl, pH 8.8, 16.6 mM ammonium sulfate, 6.7 mM magnesium chloride, 5 mM β-mercaptoethanol, 1 mg/mL proteinase K) overnight at 56 °C. Afterwards, the mixture was heated to 95 °C for 20 min, centrifuged shortly, and 5 µl of the supernatant were used for PCR with the corresponding screening primers (see Appendix A).

### 4.3. Analysis of Gene Expression on mRNA Level

RNA was extracted using the PeqGOLD total RNA kit (#732–2871, Peqlab/VWR Life Science, Bruchsal, Germany). For the analyses in Figure 1 and Figure 2, and Appendix A, cDNA synthesis and qRT-PCR were carried out as described [64]. All other gene expression analyses were performed with the qScript™ Flex cDNA Kit (#95049; Quantabio, Beverly, MA, USA) for cDNA preparation, and the PerfeCTa^®^ SYBR^®^ GreenSuperMix (#95054; Quantabio) for qRT-PCR. Here, the reaction was carried out on a CFX384 Touch Real-Time PCR Detection System (BioRad Laboratories, Hercules, CA, USA). Data were normalized using *GAPDH* expression values and relative expression was calculated using the 2^−ΔCt^ method. All primers were designed using Primer3Plus [82] and are listed in Appendix A.

Biotinylated cRNA for microarray experiments was prepared with the Ambion MessageAmp kit for Illumina arrays according to the manufacturer’s protocol. cRNA was hybridized to Illumina HumanHT12-v4 BeadChips (Illumina, München, Germany) following the manufacturer’s protocol. Raw microarray data were processed by using the Bioconductor R package beadarray and subsequently quantile normalized together. Illumina Probes were mapped to Entrez IDs using the Illumina Human v4 annotation data (Version 1.26) [83] from Bioconductor. If several probes mapped to the same Entrez ID, the one having the largest interquartile range was retained. Differentially expressed genes (DEGs) were determined via the limma package from R/Bioconductor [84]. Genes were considered significant with an adjusted *p*-value threshold at 0.05 (Benjamini-Hochberg). Microarray data are available in Gene Expression Omnibus (GEO) repository under accession number GSE143297.

### 4.4. Gene Set Enrichment Analysis

Fisher’s exact test was used to retrieve from the lists of regulated genes the enriched terms and pathways from the Gene Ontology collection (biological process) [85] and the Consensus database [86]. Significance threshold was set to adjusted *p*-value below 0.05.

### 4.5. TCGA Database

The TCGA database was accessed on 7 October 2019 via the TCGAbiolinks package [87] and 521 COAD RNAseq V2 data sets were downloaded for analyses. From these samples, 88 were filtered out due to missing survival annotation or non-colon primary site. The CMScaller package [88] was used to define CMS subtype for each TCGA sample. The Kmeans clustering method was used to divide the samples into two groups based on the expression of BMP signature genes. A package for survival analysis [89] was used for generating the survival curves and for assessing *p*-values.

### 4.6. Immunoblotting and Protein Extraction

Preparation of protein solutions through whole-cell lysis and nuclear extraction, western blotting, and detection of antibody:antigen complexes were performed as described previously [64]. Nuclear extracts were used for the detection of FOXA1, FOXA2, ID2, ZEB1, and GSK3β. All antibodies used are listed in Appendix A. Of note, the FOXA2 antibody used was previously shown to be cross-reactive with FOXA1 [64]. As the antibodies for pSMAD1/5/8 and SMAD1/5/8, as well as FOXA1 and FOXA2 were generated from the same species, membranes were stripped in-between detections as follows. After applying stripping solution (50 mM Tris/HCl pH 6.8, 0.7% β-mercaptoethanol, 2% SDS) for 30 min at 50 °C, membranes were washed three times with TBS-T, and re-incubated with the next primary antibody. All western blots were repeated three times with similar results. A compilation of uncropped immunoblots for all figures including densitometry readings is shown in Appendix A. The intensities of bands were determined using ImageJ.

### 4.7. Luciferase Reporter Gene Assay

Luciferase reporter gene assays were carried out as previously described [19], with the exception that 300 ng of Firefly Luciferase vector and 30 ng of Renilla Luciferase vector were used for transfection. The BMP reporter plasmid (pGL3 BRE Luciferase) with wt SMAD-binding elements (SBE) was a gift from Martine Roussel and Peter ten Dijke (Addgene plasmid # 45126 [90]; RRID: Addgene_45126). For generation of the SBE-mutated vector, a single-stranded, palindromic BRE-SBEmut oligonucleotide (5′-ctagcTCACTCCGTTACTCGCCAGGACGGGCTGTCAGGCTGGCGCCGCGGCGCCAGCCTGACAGCCCGTCCTGGCGAGTAACGGAGTGAg-3′; underlined parts denote the SBEs with the introduced mutations in italics, lowercase letters represent NheI restriction site overhangs) was self-annealed and subsequently cloned into the pGL3-Luc backbone previously linearized with NheI.

### 4.8. Spheroid Invasion Assay

Spheroid invasion assays with LS174T derivatives were performed as previously described [19]. Spheroid invasion assays with MCF10A were performed similarly, except that cells were pre-induced with the respective substances for 4 days before starting the assay. Pictures were then taken 10 days after embedding. If required, the medium was refreshed every 3 days. Invaded single cells were manually counted using ImageJ.

### 4.9. Anoikis Resistance Assay

Anoikis resistance assays were carried out by adapting a previously published protocol [27]. Briefly, 1500 cells were seeded in 96-well plates of normal tissue culture grade enabling adhesive growth and on plates with a cell-repellent surface (both by Greiner Bio-one, Kremsmünster, Austria). After overnight incubation, cells were treated with substances for 72 h and 96 h and cell viability was determined by performing CCK8 assay (ENZO Life Sciences, Farmingdale, NY, USA) according to manufacturer’s instructions. All experiments were performed in technical duplicates.

### 4.10. Software and Statistical Analyses

Figures were prepared using GraphPad Prism (San Diego, CA, USA) and Canvas X (Plantation, FL, USA). RNA analysis was performed using R (R Core Team, 2019, version 3.6.1). Statistical analyses of qRT-PCR data and spheroid invasion assay were carried out in GraphPad Prism. If not mentioned otherwise in the respective figure legends, unpaired two-tailed student’s *t*-test was routinely used to determine significant differences between two samples. Asterisks mark statistical significance according to the following *p*-value thresholds: * *p* < 0.05, ** *p* < 0.01, *** *p* < 0.001. A summary of all statistical analyses carried out, including the obtained *p*-values, can be found in Appendix A.

## 5. Conclusions

In summary, our study demonstrates that SNAIL1 employs canonical BMP signaling for EMT execution in colorectal cancer, and likely in a variety of additional pathophysiological and physiological conditions where EMT occurs. Our findings therefore reevaluate the connection between BMP signaling and EMT and substantially extend our knowledge about the mechanistic routes through which EMT can be implemented. Thus, they imply novel strategies for the therapy of EMT-associated malignancies.

## Figures and Tables

**Figure 1 cancers-12-01019-f001:**
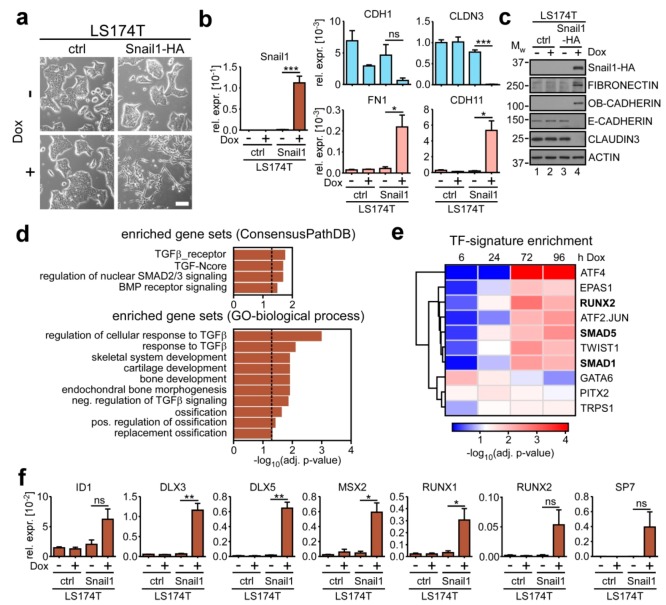
SNAIL1 induces bone morphogenetic protein (BMP) target genes during epithelial-mesenchymal transition (EMT) in colorectal cancer cells. (**a**) Representative phase contrast images of LS174T-ctrl and LS174T-Snail1-HA cells. Cells were left uninduced or were treated with 0.1 µg·mL^−1^ Dox for 72 h. Scale bar: 100 µm. (**b**) qRT-PCR analyses of mRNA expression in LS174T-ctrl and LS174T-Snail1-HA cells. Where indicated, cells were treated with 0.1 µg·mL^−1^ Dox for 72 h. Shown is the mean + SEM; *n* = 3. Rel. expr.: relative expression normalized to that of *GAPDH*. ns: not significant. *: *p* < 0.05, ***: *p* < 0.001. (**c**) Western blot analyses of whole-cell lysates. Names of detected proteins are indicated on the right. Cells received 0.1 µg·mL^-1^ Dox or were left untreated. Positions of molecular weight (M_W_) standards in kDa are given on the left. Detection of ACTIN was used as control for equal loading. As not all proteins could be analyzed on the same membrane, only one representative loading control is shown for reasons of simplicity. All corresponding loading controls for the images depicted can be found in Appendix A. (**d**) Gene set enrichment analysis (GSEA) of the genes upregulated by Snail1-HA after 72 h of Dox administration. A selection of significantly enriched gene sets is shown. Plotted are the negatives of the log_10_ of the adjusted (adj.) *p*-values. Vertical dotted line indicates the applied cutoff of adj. *p*-value ≤ 0.05. A complete list of terms analyzed can be found in Appendix A. (**e**) Top ten enriched transcription factor (TF)-signatures in the genes deregulated upon induction of Snail1-HA in LS174T after Dox treatment for different time spans. Hours of Dox administration are indicated on top. Enrichment scores are plotted as negatives of the log_10_ of the adjusted (adj.) *p*-values. A complete list of enriched TF-signatures is given in Appendix A. (**f**) qRT-PCR analyses of mRNA expression in LS174T-ctrl and LS174T-Snail1-HA cells. Where indicated, cells were treated with 0.1 µg mL^−1^ Dox for 72 h. Shown is the mean + SEM; *n* = 3. Rel. expr.: relative expression normalized to that of *GAPDH*. ns: not significant. *: *p* < 0.05, **: *p* < 0.01.

**Figure 2 cancers-12-01019-f002:**
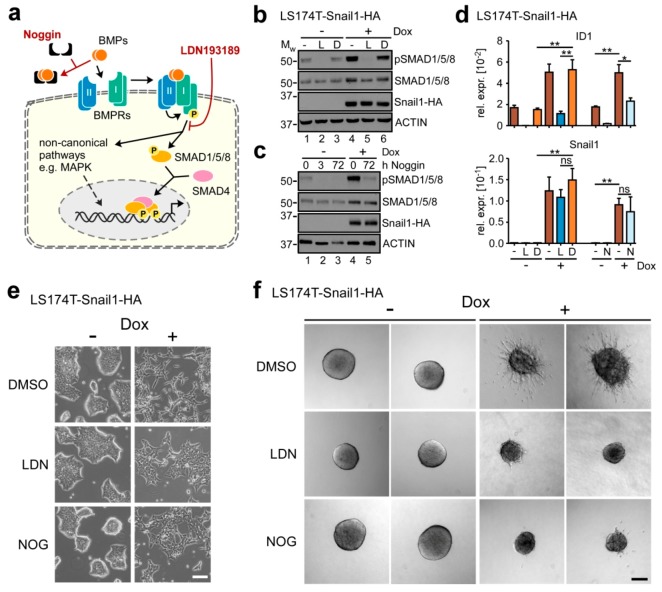
Inhibition of the BMP pathway strongly impairs the SNAIL1-induced EMT in colorectal cancer cells. (**a**) Schematic depiction of the BMP signaling pathway. The two inhibitors Noggin and LDN193189 interfere with signal transduction by sequestering BMP ligands and inhibiting BMP type I receptor A (ALK3), respectively. (**b**) Western blot analyses of whole-cell lysates. Names of detected proteins are indicated on the right. Cells were left uninduced or were treated with 0.1 µg·mL^−1^ Dox and 50 nM LDN193189 (L), or DMSO (D) for 72 h. Positions of molecular weight (M_W_) standards in kDa are given on the left. Detection of ACTIN was used as control for equal loading. (**c**) Western Blot analyses of whole-cell lysates. Names of detected proteins are indicated on the right. Cells were left uninduced or were treated with 0.1 µg·mL^−1^ Dox and 100 ng·mL^−1^ Noggin for the indicated time spans. Positions of molecular weight (M_W_) standards in kDa are given on the left. Detection of ACTIN was used as control for equal loading. (**d**) qRT-PCR analyses of mRNA expression in LS174T-Snail1-HA cells. Where indicated, cells were treated with 0.1 µg·mL^−1^ Dox, 50 nM LDN193189 (L), DMSO (D), or 100 ng·mL^−1^ Noggin (N) for 72 h. Shown is the mean+SEM; *n* = 3. Rel. expr.: relative expression normalized to that of *GAPDH*. ns: not significant. *: *p* < 0.05, **: *p* < 0.01. (**e**) Representative phase contrast images of LS174T-Snail1-HA cells treated with 0.1 µg·mL^−1^ Dox and DMSO, 50 nM LDN193189 (LDN), or 100 ng·mL^−1^ Noggin (NOG) for 72 h as indicated. Scale bar: 100 µm. (**f**) Spheroid invasion assay of LS174T-Snail1-HA cells treated with 0.1 µg·mL^−1^ Dox and DMSO, 50 nM LDN193189 (LDN), or 100 ng·mL^−1^ Noggin (NOG) for 96 h as indicated. Two representative spheroids are shown for each condition. Scale bar: 200 µm.

**Figure 3 cancers-12-01019-f003:**
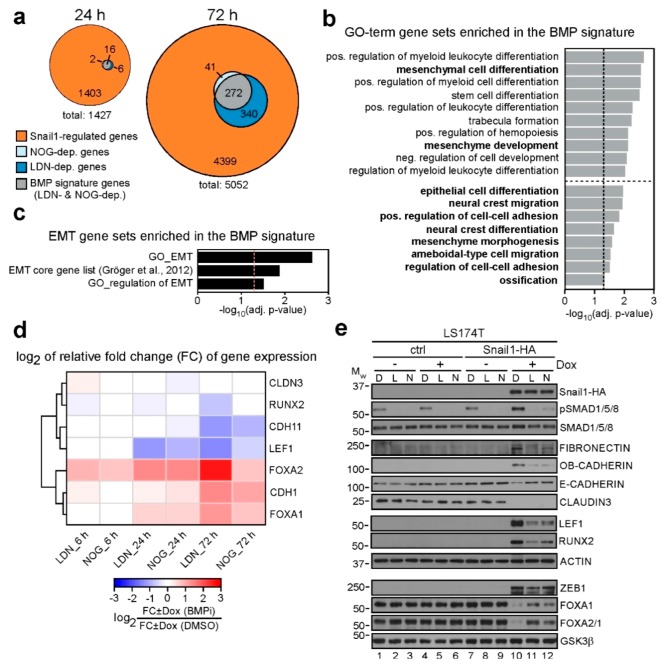
Defining a high-confidence BMP-dependent gene expression signature in the Snail1-induced EMT transcriptome. (**a**) Euler diagrams displaying the numbers of Snail1-regulated genes whose regulation is impaired by BMP inhibitor treatment at 24 and 72 h of Snail1-induction. When comparing the fold changes of gene regulation in the scenario of BMP inhibitor treatment to DMSO treatment, an adjusted *p*-value < 0.05 was used as a cutoff to determine BMP-dependent genes. Grey overlaps represent the BMP signature genes. Time points indicate duration of induction with 0.1 µg·mL^−1^ Dox. (**b**) Gene set enrichment analysis (GSEA) showing significantly enriched gene sets in the BMP signature genes. Shown are the top ten significantly enriched gene sets, as well as, separated by the horizontal dotted line, additional selected significant terms. Plotted is the negative log_10_ of the adjusted *p*-value. Vertical dotted line indicates the cutoff of adj. *p*-value ≤ 0.05. A complete list of terms analyzed can be found in Appendix A. (**c**) GSEA results of EMT-related gene sets significantly enriched in the BMP signature genes. Vertical dotted line indicates the applied cutoff of adjusted (adj.) *p*-value ≤ 0.05. A complete list of terms analyzed can be found in Appendix A. (**d**) Heatmap showing the impact of BMP inhibitor treatment on the regulation of genes of interest in LS174T-Snail1-HA at different time points of induction with 0.1 µg·mL^−1^ Dox. Plotted is the log_2_ of the relative fold change of gene expression under BMP inhibitor treatment. Relative fold change was calculated by dividing gene expression differences induced by Snail1-HA in the scenario of BMP inhibition by the differences evoked by Snail1-HA under control conditions. Inhibitors used and durations of administration are indicated on the x-axis. As their expression was not properly detected in the microarray experiment, *FN1* and *ZEB1* are not shown. (**e**) Western blot analyses of proteins indicated on the right. Cells were left uninduced or were treated with 0.1 µg·mL^−1^ Dox and DMSO (D), 50 nM LDN193189 (L), or 100 ng mL^-1^ Noggin (N) for 72 h as indicated. Positions of molecular weight (M_W_) standards in kDa are given on the left. Detection of ACTIN and GSK3β was used as control for equal loading. Proteins were detected using whole-cell lysates, except for FOXA1, FOXA2, ZEB1 and GSK3β for which nuclear extracts were used. As not all proteins could be analyzed on the same membrane, only one representative loading control is shown for the different extraction methods for reasons of simplicity. All corresponding loading controls for the images depicted can be found in Appendix A.

**Figure 4 cancers-12-01019-f004:**
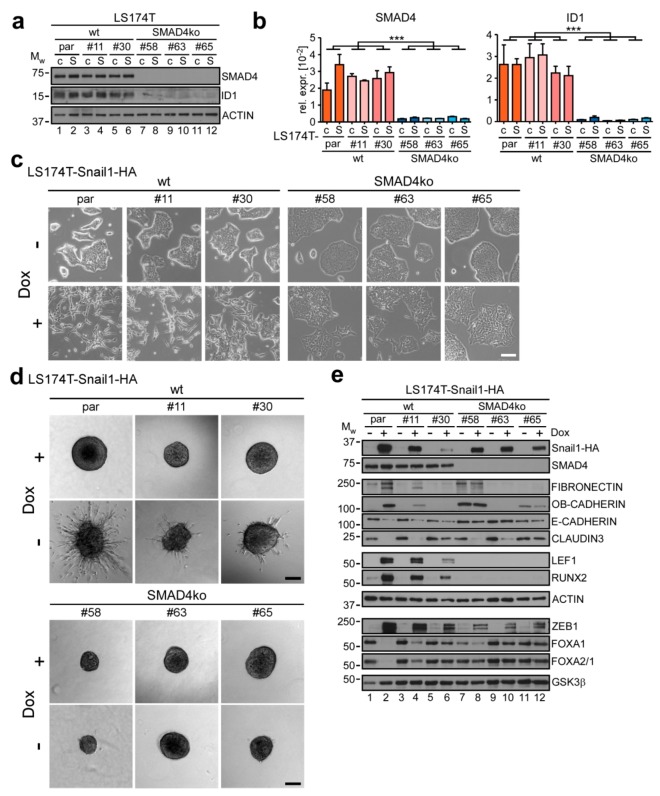
SMAD4 is required for SNAIL1-induced EMT in colorectal cancer cells. (**a**) Western blot analyses of whole-cell lysates. Names of detected proteins are indicated on the right. Positions of molecular weight (M_W_) standards in kDa are given on the left. Detection of ACTIN was used as control for equal loading. par: parental, c: control, S: Snail1-HA. (**b**) qRT-PCR analyses of mRNA expression in LS174T parental (par) cells and single cell clones reconstituted with inducible control (**c**) or Snail1-HA (S) expression vectors. Statistical significance was analyzed by first testing for differences among the three wt (par, #11, #30) and the three *SMAD4* ko (#58, #63, #65) clones by one-way ANOVA. As there were no significant intra-group differences for both genes, values for each clone were averaged and inter-group differences between wt and *SMAD4* ko were determined using unpaired two-tailed student’s *t*-test. Shown is the mean + SEM; *n* = 3. Rel. expr.: relative expression normalized to that of *GAPDH*. ***: *p* < 0.001. (**c**) Morphology of LS174T parental (par) cells and single cell clones reconstituted with inducible Snail1-HA expression vector. Cells were left uninduced or received 0.1 µg·mL^−1^ Dox for 72 h. Scale bar: 100 µm. (**d**) Spheroid invasion assay of LS174T parental (par) cells and single cell clones reconstituted with an inducible expression vector for Snail1-HA. Cells were left untreated or received 0.1 µg·mL^−1^ Dox for 72 h. One representative spheroid is shown per condition. Scale bar: 200 µm. (**e**) Western blot analyses of proteins indicated on the right. Cells were left untreated or received 0.1 µg·mL^−1^ Dox for 72 h. Positions of molecular weight (M_W_) standards in kDa are given on the left. Detection of ACTIN and GSK3β was used as control for equal loading. Proteins were detected using whole-cell lysates, except for FOXA1, FOXA2, ZEB1 and GSK3β for which nuclear extracts were used. As not all proteins could be analyzed on the same membrane, only one representative loading control is shown for the different extraction methods for reasons of simplicity. All corresponding loading controls for the images depicted can be found in Appendix A.

**Figure 5 cancers-12-01019-f005:**
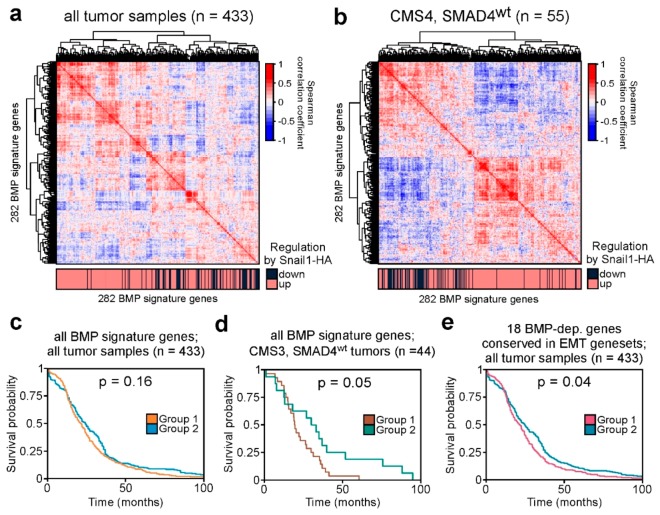
BMP signature gene expression is recapitulated in human colorectal cancer tumors and can be predictive for patient survival after tumor stratification. (**a**), (**b**) Correlation maps showing the mutual correlation of expression levels of all 282 BMP signature genes in transcriptomes of colorectal cancer (CRC) samples. Genes were clustered by unsupervised hierarchical clustering based on the Euclidean distance. In (**a**), all CRC samples available from TCGA were used. For (**b**), only the subset of tumors that classified as consensus molecular subtype (CMS) 4 and without mutations in *SMAD4* (*SMAD4*^wt^) was considered. The color bar on the bottom of each plot indicates whether a gene is up- or downregulated by Snail1-HA in LS174T cells. (**c**–**e**) Kaplan–Meier curves indicating survival probability of colorectal cancer patients. Patients were separated into two groups based on the expression of BMP signature genes in the tumors. In (**c**), all tumor samples and all BMP signature genes were used for the analysis. In (**d**), only the CMS3, *SMAD4*^wt^ tumors and, in (**e**), only 18 of the BMP signature genes conserved in significantly enriched EMT-related gene sets (see Figure 3c and Appendix A) were considered. The corresponding clustering heatmaps are shown in Appendix A. Log-rank test was applied to determine the significance of differences in survival probability; *p*-values are indicated.

**Figure 6 cancers-12-01019-f006:**
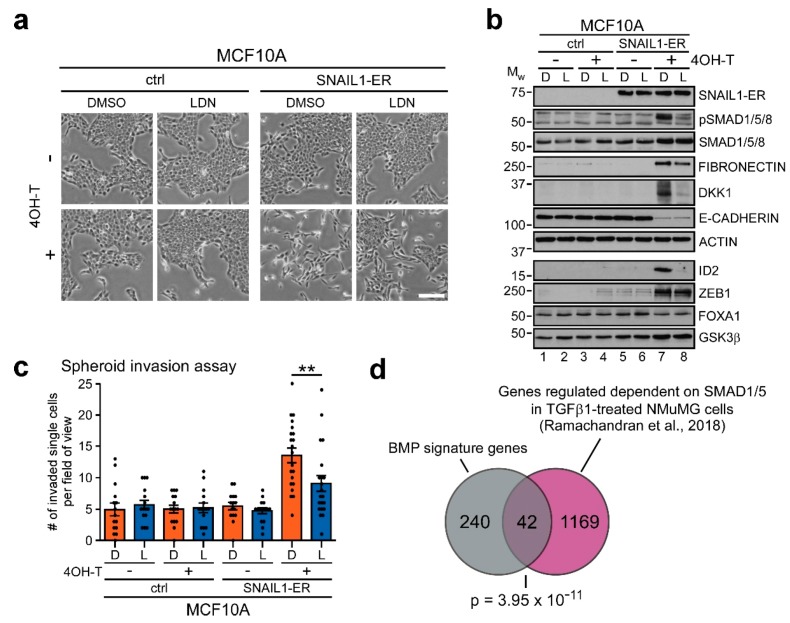
BMP signature genes are also deregulated BMP-dependently in mammary epithelial cells. (**a**) Representative phase contrast images of MCF10A-ctrl or MCF10A-SNAIL1-ER cells treated with ethanol (−) or 100 nM 4-hydroxytamoxifen (4OH-T) and DMSO, or 50 nM LDN193189 (LDN) for 10 days as indicated. Scale bar: 200 µm. (**b**) Western blot analyses of proteins indicated on the right. Cells were treated with ethanol (−) or 100 nM 4-hydroxytamoxifen (4OH-T) and DMSO (D), or 50 nM LDN193189 (L) for 10 days as indicated. Positions of molecular weight (M_W_) standards in kDa are given on the left. Detection of ACTIN and GSK3β was used as control for equal loading. Proteins were detected using whole-cell lysates, except for FOXA1, ID2, ZEB1 and GSK3β for which nuclear extracts were used. As not all proteins could be analyzed on the same membrane, only one representative loading control is shown for the different extraction methods for reasons of simplicity. All corresponding loading controls for the images depicted can be found in Appendix A. (**c**) Quantification of spheroid invasion assays with MCF10A-ctrl and MCF10A-SNAIL1-ER cells. Cells were treated with ethanol (−) or 100 nM 4-hydroxytamoxifen (4OH-T) and DMSO (D), or 50 nM LDN193189 (L) for 14 days as indicated. After that, cells were imaged and invaded single cells were counted in ≥15 fields of view per condition, which were obtained from three biological replicates. Representative fields of view with examples of single cell quantification are shown in Appendix A. Shown is the mean ± SEM as well as all individual data points obtained. Data were tested for normality using Shapiro–Wilk tests (*p* = 0.018 for sample 8, indicating non-normal distribution). Significance was therefore determined using two-tailed Mann–Whitney U tests. **: *p* < 0.01. (**d**) Venn diagram showing the overlap of BMP signature genes with all genes that are regulated by TGFβ1 in NMuMG cells in a SMAD1/5-dependent manner [39]. Shared genes and details of the analysis are listed in Appendix A. The significance of enrichment was determined by hypergeometric test; the *p*-value is indicated.

**Figure 7 cancers-12-01019-f007:**
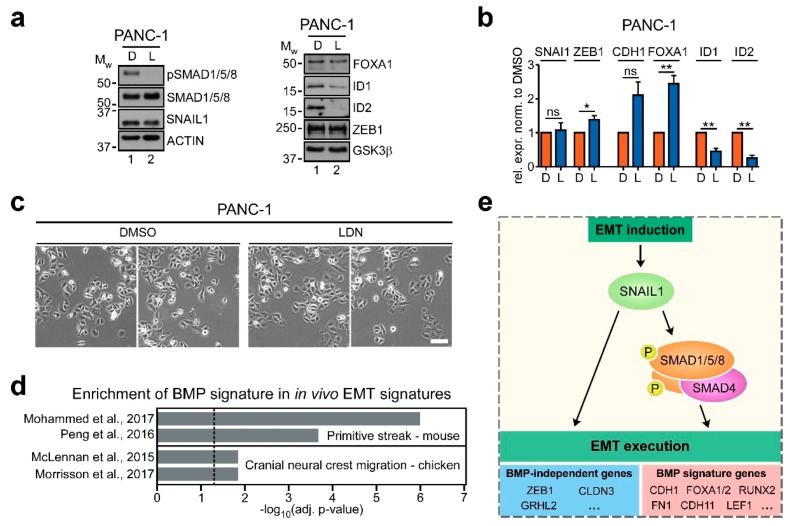
BMP pathway activity partially sustains a mesenchymal state of pancreatic cancer cells and BMP signature gene expression occurs in developmental instances of EMT. (**a**) Western blot analyses of proteins indicated on the right. PANC-1 cells were treated with DMSO or 50 nM LDN193189 (LDN) for 72 h as indicated. Positions of molecular weight (M_W_) standards in kDa are given on the left. E-CADHERIN could not be detected. Detection of ACTIN and GSK3β was used as control for equal loading. Proteins were detected using whole-cell lysates, except for FOXA1, ID1, ID2, ZEB1 and GSK3β for which nuclear extracts were used. (**b**) qRT-PCR analyses of mRNA expression in PANC-1 cells treated with DMSO or 50 nM LDN193189 (LDN) for 72 h as indicated. Significance was determined using two-tailed one sample t-test. Shown is the mean+SEM; *n* = 4. Rel. expr.: relative expression normalized to that of GAPDH. ns: not significant. *: *p* < 0.05, **: *p* < 0.01. (**c**) Phase contrast images of PANC-1 cells treated with DMSO or 50 nM LDN193189 (LDN) for 72 h as indicated. Two representative fields of view are shown for each condition. Scale bar: 100 µm. (**d**) Gene set enrichment analysis (GSEA) of BMP signature genes in publicly available sets of genes regulated in EMT-associated processes in different organisms in vivo. Only genes upregulated by Snail1-HA in LS174T cells were considered. Source publications of the analyzed gene sets are given on the left. Plotted is the negative log10 of the adjusted (adj.) *p*-value. Vertical dotted line indicates the significance threshold of adjusted (adj.) *p*-value ≤ 0.05. A detailed list of GSEA results is given in Appendix A. (**e**) Model depicting the proposed role of BMP signaling in EMT execution downstream of SNAIL1. Upon induction of EMT and upregulation of SNAIL1, BMP pathway-associated SMAD transcription factor complexes are activated and control a subset of critically EMT-relevant genes. Examples for such BMP signature genes are shown. Note that SNAIL1 regulates additional epithelial and mesenchymal marker genes independently of BMP pathway activity.

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
