# Peer review of "Canonical BMP Signaling Executes Epithelial-Mesenchymal Transition Downstream of SNAIL1"

_cancers, 2020, doi:10.3390/cancers12041019_

Round 1

Reviewer 1 Report

The authors improved considerably the manuscript and I am looking forward to see it published. The data is extremely relevant and presented in a way that is accessible to other researcher, namely providing all the supplementary data.

Author Response

We are grateful to the reviewer for his/her approving and favourable comments.

Reviewer 2 Report

The authors have performed additional experiments that were strongly suggested in the previous critique, for which they should be commended.

In their rebuttal letter, they state that the results of their assays for anoikis and changes in Grainyhead expression yielded negative results.

They have now chosen not to disclose these results in their manuscript.

Scientific rigor dictates disclosing both positive and negative results. To hide their negative results would be inadvisable from this point of view.

The authors need to at least state, as "data not shown" or show (in supplemental online data) their negative results on both anoikis and GRHL2, with appropriate explanation of why these experiments were attempted. 

Author Response

The authors have performed additional experiments that were strongly suggested in the previous critique, for which they should be commended.

In their rebuttal letter, they state that the results of their assays for anoikis and changes in Grainyhead expression yielded negative results.

They have now chosen not to disclose these results in their manuscript.

Scientific rigor dictates disclosing both positive and negative results. To hide their negative results would be inadvisable from this point of view.

The authors need to at least state, as "data not shown" or show (in supplemental online data) their negative results on both anoikis and GRHL2, with appropriate explanation of why these experiments were attempted.

Response: As requested, the results of the anoikis resistance test and the GRHL2 expression analyses are now part of our manuscript.

We mention anoikis resistance as one of the EMT-associated features which were examined under BMPi (Results, lines 159-170) and present the results in the newly added Figure S3. As a consequence the Materials and Methods section was expanded, and the new reference 27 was added.

GRHL2 was included in the list of EMT-relevant genes which we analyzed (Results; lines 242-249, line 456). The relevant qRT-PCR results for LS174T cells are shown in the expanded Figure S5, panel c. Again, additional references (refs. 31-33) were added to provide background information for GRHL2.

Details for the newly added experiments and all the associated changes in the various parts of the manuscript, figures, and supplementary material are listed below.

This manuscript is a resubmission of an earlier submission. The following is a list of the peer review reports and author responses from that submission.

Round 1

Reviewer 1 Report

This is a well-written and carefully performed study examining the important role of BMP signaling downstream of Snail in enforcing EMT.

I have only a couple of criticisms:

a. I am not aware of any tumor cells that have only Snail over-expression and no other (epi)genetic changes. So the main experimental system is a bit artificial. May I suggest the converse approach:  knocking down Snail expression in an EMT-like cell line, and assaying for BMP signaling.  

b.  The authors should really look at GRHL2 expression because it is a robust marker of epithelial phenotype and it enforces the latter and it is down-regulated during EMT.

c.  The authors should really look at anoikis-sensitivity or resistance as it is an accepted hallmark of epithelial vs. mesenchymal phenotypes.

Reviewer 2 Report

This study addresses with elegance, and with a wide variety of complementary approaches the mechanisms involved in the so-called “Epithelial-Mesenchymal-Transition” (EMT), essential in development but also involved in cancer progression. The authors explain that, while some information exists regarding the triggering of the EMT, little is know about its execution. They demonstrate that one of the Tfs involved in EMT triggering, SNAIL1, relies on the canonic SMAD-dependent BMP signaling. In addition they define a BMP-dependent signature involved in EMT that is recapitulated in human Colorectal Carcinoma patients, and with a prognostic value for cancer survival. They also show that these BMP-related genes are regulated during embryonic development, suggesting their involvement in EMT. An important aspect of this work is that uncovers a mechanism that might be common to distinct EMT processes. In their functional assays, the authors used colon and breast in vitro models, allowing SNAIL1 overexpression, and found BMP-signature genes deregulated in both. Furthermore they also used data from animal structures, in which EMT is expect to occur, and they also found active BMP-signature genes.

In summary, I think that the authors present very interesting data that should be published, clearly adding to the current state of knowledge on the mechanisms of execution of the EMT, independently of the tissue/cellular context.

I provide some suggestions bellow

I missed a summary figure in the end. Line 56: “orchestrating EMT in cooperation with and downstream of EMT-TFs. I found this sentence confusing and not straight to the point. Line 66: “BMP signaling events have central functions in development and adult organs”. I think it should be: “BMP signaling events have central functions in development and in adult organs”. Line 77: First sentence of the paragraph needs revision. Figure 1, 2 and 4: no statistical analyses were done to compare the RT-qPCRs results.